# The Catholic Charismatic Movement in Global Pentecostalism

**Enzo Pace** 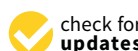

Galilean School of Higher Education, University of Padua, Via Cesarotti 12, 35123 Padova, Italy;
vincenzo.pace@unipd.it

**Abstract:** This article deals with Catholic Charismatics in Italy. The brief description of the case study gives a chance to make some more general comments on what is happening under the sacred canopy of Global Catholicism where the Spirit blows, and furthermore in relation with so-called Global Pentecostalism. In other words, my working hypothesis includes the following statements: (a) Catholic Pentecostalism constitutes a variant of a more global phenomenon, which seems to challenge the organizational model of historic Christian churches. (b) The study of the Italian case is interesting because its story shows the extent to which Pentecostalism questions the Roman form of Catholicism. Elsewhere in the world, the development of the phenomenon has not encountered the same difficulties as it did in Italy. Indeed, in some cases (Brazil and the Philippines), it has been supported and accepted as a sign of new religious vitality. From this point of view, Rome is relatively far away. The Roman–Tridentine model governed by the clergy resists in Italy, while it appears weaker where the Spirit blows wherever it wants. The Charismatic movement was gradually brought back to the bed of ecclesial orthodoxy after a long persuasive work carried out by bishops and theologians towards the leaders of the movement itself. However, despite this ecclesification/clericalization process, the charismatic tension remains, and the expectation for a pneumatic church constitutes an implicit form of criticism of the Roman form of Catholicism.

**Keywords:** Catholic Charismatic movement; Italy; Global Pentecostalism

## 1. Introduction

The Catholic Charismatic movement (CCR) is part of the vast and highly articulated reality that, for the sake of convenience, scholars call Global Pentecostalism as a sociolinguistic shortcut (Miller and Yamamori 2007). According to reliable estimates, more than 500 million people are involved in the many varied Pentecostal and Charismatic groups and churches (Barrett et al. 2001; Pew Research Center 2006, 2011; Csordas 2007; Nogueira-Godsey 2012). The differentiated models of aggregation have also changed during the so-called three waves of Pentecostalism (Hollenweger 2005; Anderson and Bergunder 2010): the first was historical or classical Pentecostalism (1900–1939); the second was the Neo-Pentecostalism of the 1960s and 1970s; and the third began in the early 1980s, taking the form of what I propose to call the charismatic enterprise. The three waves do not only mark a diachronic succession. They represent a process of differentiation within the field of Global Pentecostalism (Anderson 2013). The theological formula according to which the Spirit blows where it will in sociology means that Global Pentecostalism shares the same feeling, albeit in diverse forms, animating different places without real territorial boundaries (Ericksen et al. 2019). The religious imagination—imagining the Kingdom of God—overrides the orderly systems of belief controlled by the Protestant and Catholic Churches. Since the 1980s, Neo-Pentecostal churches have liquidated not only the heritage of the Reformation, but also the theological perspective inaugurated by the first wave of Pentecostalism.

The CCR is one of several movements born in the period immediately following the Second Vatican Council. They interpreted the widespread reform expectations of the Catholic Church which had been expressed since the early 1950s in different areas of the Catholic world. I only remember other movements such as the neo-catechumenal movement, the Cursillos of Christianity, Communion and Liberation (CL) and the basic communities: Catholic movements often born on the impetus of lay people (except for CL, founded by a Milanese priest, Don Luigi Giussani) (Faggioli 2016).

This article presents a study on the Catholic Pentecostal movement in Italy (the official name of the Italian CCR is "Rinnovamento nello Spirito"—Renewal in the Spirit): from its origins to its definitive recognition by the authorities of the Catholic hierarchy, its diffusion and its organizational model. The focus of the analysis, however, is broader; it moves from the particular towards the general, where by general I mean not only the Catholic Charismatic movement spread in many other countries in the world, but the modern form of Pentecostalism that started in the American evangelical environment over a hundred years ago. In other words, I am assuming that the Catholic Pentecostal movement is one of the various expressions and manifestations of a much more complex and articulated field of religious forces.

Pentecostalism, worldwide, can be seen as a religious field, relatively autonomous, differentiated and articulated, where the symbolic representation of the Kingdom has favoured the experimentation of a spiritual and organizational model explicitly or implicitly in conflict or in tension with the various models elaborated in the history of Christianity by the main historical Churches (Catholic, Reformation and the varied and complex world of Orthodoxy).

Bourdieu's notion of a religious field (Bourdieu 1971) continues to be useful, although it should be integrated with another notion that comes from Jacob Taubes' studies (Taubes 1994) on Christian eschatology. Bourdieu connects the idea of the field with the "class struggle" between those who hold power and knowledge about symbolic goods (symbolic representations, to be precise, referring to the 1971 article), on the one hand, and those who do not have them, on the other. According to French scholar, the religious conflict implies a rate of symbolic violence, since specialists in sacred things try to impose on the non-specialists a specific religious representation of the world, the cosmic and social order. So, the symbolic representations, for Bourdieu, have to do with the representations of the social order. The religious field is a system of believing governed by a regime of monopoly, oligopoly or pluralism of faiths. The more a monopoly prevails, the more a belief system does not tolerate the emergence of forms of belief and practices that challenge the monopoly regime itself. This happens today, as in the past, in Catholicism, when a lay-led movement questions clerical power and, ultimately, doubts the Catholic principle according to which (ecclesiastical) authority guarantees truth.

If we add to Bourdieu's approach the ideas elaborated between 1945 and 1947 by Jacob Taubes, a rabbi, philosopher and also, for a short season of his intellectual life, a sociologist of religion (when Gershom Scholem called him to teach this discipline at Hebrew University of Jerusalem between 1951 and 1953), the point of view that I intend to take in the analysis of Catholic Pentecostalism is more appropriate. When we speak, in fact, of modern Pentecostalism, in all its variants (Catholic, Evangelical, Orthodox) and its variations on a theme (from classical Pentecostalism to neo-Pentecostalism and, finally, to the charismatic enterprise), we are talking about the imagination of the Kingdom by various collective movements that refer to the Christian message of Salvation at the End of Times. They are socio-religious actors who, precisely, mobilize crucial symbolic resources that belong to the Christian message and try to keep together the belief in the promise of the Kingdom and the practical forms of spirituality, rituality and organization which are considered more adequate to faithfully reflect this eschatological expectation.

Therefore, studying Catholic Pentecostalism means taking seriously into account that it is as an articulation of a wider Christian–Pentecostal field. Without this broader look, in my opinion, there is a risk of not grasping the religious innovation that overall modern Pentecostalism has proposed and is proposing. In this way, it is possible to measure the impact of the Pentecostal phenomenon on the historical Churches in the religious field. It is true for those churches that have hitherto acted in

a monopoly position, but because of the change of the historical and social context no longer seem guaranteed; in particular, in those societies that became highly religious due to the migrations and to the new forms of belief by choice and no longer by tradition. It also applies to the Reformation's Churches, aware of their structural precariousness (Willaime 1999), unable to appeal to any Primacy of Peter. They look with a theological suspicion all of the enthusiastic manifestations and belief in miracles professed by Pentecostals. However, for the Catholic Church and for the Reformation's Churches, the Pentecostal movements are not only competitors in contemporary free market of salvation goods, but also producers of a new Christian semantics. A new wine in old wineskins. It means that the Pentecostal social actors no longer trust the words used by the historical Churches about the Kingdom of God.

The key word, for example, "Baptism *with* the (Holy) Spirit" that occurs in Pentecostalism is an ancient notion (although polysemic and controversial in the history of Christianity) in the official theologies of the major historical churches. In modern Pentecostalism, this key word is covered with new meanings and, depending on the contexts in which movements arise by measuring themselves with the constraints of the surrounding cultural environment and the main dominant religious institutions, is declined in a different way. In the Catholic Church, for example, the formula that ultimately prevailed in Roman ecclesiastical circles to name the Catholic Pentecostal movement was "Baptism *in* the Spirit". With this formula, Catholic theologians who joined the CCR explained that the Holy Spirit places a believer into permanent union with Christ and with other believers in the Body of Christ, represented by the Church; experiencing baptism in the Holy Spirit is an exhortation and spiritual effort to adhere to the mystical body of Christ, the Church. In sociological terms, it is a way for the Catholic hierarchy to reduce the socio-religious complexity with which Pentecostalism challenges the spiritual, liturgical and organizational model of the Roman–Tridentine form of Catholicism. It is a way to mark a distance with both classical and new Pentecostalism; both the Spirit blows where it will and reveals itself with its gifts to anyone who entrusts it. According to the Pentecostals' semantics, Baptism with the Spirit is instead an empowering experience for individuals and a community.

All this explains the rationale of the article. I am moving to tell the story of the Catholic Pentecostal movement in Italy (from its origins to its official recognition and its roots in the whole national territory), putting it then into perspective and in comparison both with what has happened and happens in other countries with a long Catholic tradition, and as regards global Pentecostalism. Catholic Pentecostalism is differentiated internally. This differentiation can be explained on the basis of both extrinsic and intrinsic criteria: extrinsic, the relative proximity/distance from Rome, understood not only as the Center of Catholicity but also as a universal and only model of how a church should be; intrinsic to the Christian message: what is the organizational pattern most consistent with the eschatological waiting for the Kingdom? A pure congregationalist pattern or a federative agreement among various communities that, however, implies a minimal hierarchical apparatus?

There is a certain difference between those who, militating in a Catholic Pentecostal group (Marzano 2009), wish to maintain a filial attitude towards Holy Mother Church, and those who, while continuing to feel Catholic, do not tolerate ecclesiastical authorities' interference to limit and control the effervescence of the charisms. Therefore, some people decide to leave the Catholic Church, creating an autonomous movement, or to join the Assemblies of God or other Pentecostal and Evangelical denominations (Introvigne and Zoccatelli 2010; Pace 2013).

Framing CCR in global Pentecostalism, we can better assess the extent of the challenge that the CCR also poses to the Catholic Church, despite the process of the official institutionalization of charismatic movements completed by pope John Paul II in 1998. In a broad sense, the challenge concerns precisely the way of being a church directly experiencing the presence of the Spirit without authoritative ecclesiastical mediation (Catholicism) or the cognitive supervision of the pastors (Reformation). The first Catholic charismatic communities saw in the experience of the Spirit a way to overcome the functional separation between clergy and lay people (Pace 1983) or, at least, to live liturgically the *universal priesthood* as enunciated by the Vatican II in the 'Lumen Gentium'. In that sense, for the

charismatic movement at the *status nascendi*, the word 'church' began to take on at least two meanings: a hierarchical structure led by bishops and the Pope to whom obedience is due, and also the celestial church (more spiritual) revealed though the mystical experience of the Spirit. A light church vs. a big infrastructure of the sacred. Paraphrasing the last pages of Max Weber's Protestant Ethics (Weber 2001, p. 123), the Catholic charismatics was like wearing a light cloak to be put on the shoulders and not an iron cage that suffocates the spirit. It is not new. It is a recurrent Christian imagination that is not always realized, since it can happen that, in the passage from the ecstatic experience to the most appropriate organizational model to be adopted to be reproduced, that over time the risk of the 'routinization of charisma' is always and equally recurrently higher.

A final consideration on the choice to compare horizontally the main differentiations of the Pentecostal religious field. In social realities, where the differentiation exists (conspicuously, for example, in Brazil or the Philippines, more discreetly in some European societies, such as Belgium, France, Italy and Poland), the ethnography of various communities belonging respectively to classical Pentecostalism, neo-Pentecostalism and Catholic Pentecostal renewal allows us to observe two socio-religious effects.

The first effect is the mimetic one; this is particularly striking in those countries where the Catholic Church loses ground and some Catholic charismatic leaders tend to adopt communication styles like those of the leaders of the new Pentecostal churches. Brazil, from this point of view, is a very interesting laboratory. The second effect is the open competition by Catholic charismatics, played on the same theological terrain as the ritual performances of the neo-Pentecostal movements. In some cases, we are witnessing a pentecostalization of Catholicism thanks to emergence of very popular charismatic leaders, capable of counteracting the advance of neo-Pentecostal Christianity. We are, in this case, beyond imitation, since the copy is not always distinguishable from the original. The Philippines is the most exemplary case from this point of view.

This article is divided into three parts according to a structure of concentric circles, from the smallest to the largest. The first part is dedicated to the story of the Catholic Pentecostal movement in Italy (the CCR, called the 'Renewal in the Spirit') and, in particular, to the map of the spread not only of Catholic communities, but also to the growth of the Assemblies of God, on the one hand, and the emergence of the new Pentecostal churches of African, Latin American and Chinese origin, on the other. The second part broadens our gaze to other realities historically marked by Catholicism, where the CCR has successfully taken roots from the seventies to today. The idea is to underline the different vicissitudes of the relationships between these movements at the national level and the attitude of the Catholic hierarchy. All had to 'pass examinations' and tests of control, but in the Italian case the CCR had to undergo a long examination by theologians and custodians of Catholic doctrine before obtaining full recognition. The further away Rome they are, the less pressure the movements experienced elsewhere. In the third part we will resume the main argument, focusing on the differentiation of the Pentecostal field, in a way that is capable of satisfying a wide range of religious expectations, from the individualized need for religious self-consumption to the desire to experience directly a power (that of the Spirit), while remaining inside the reassuring walls of one's domestic Church. In the midst of these two poles, we find in the reality of global Pentecostalism a variety of ways and themes, to bring up Troeltsch's (Troeltsch 1947) attempt to combine the uniqueness and absoluteness of the message of Jesus the Christ with the plurality of spiritual, liturgical and organizational models (church-type, sect-type and *spiritualismus*-type) largely overcome by reality itself, with the emergence of a new pattern of organization: the market-oriented charismatic enterprise.

The religious imagination—imagining the Kingdom of God—overrides the orderly systems of belief controlled by the Protestant and Catholic Churches. Since the 1980s, the Neo-Pentecostal churches have liquidated not only the heritage of the Reformation, but also the theological perspective inaugurated by the first wave of Pentecostalism.

The new Churches soon learned to operate in a crowded and competitive religious field. They learned the art of communication not only vertically, with the power of the Spirit, but also and above all horizontally, in a market of salvation goods offered by a plurality of similar actors. Seen from

the supply side, the competition stimulated a religious creativity that is concerned less with the content (which is often much the same in the churches of the second wave, whatever their size), and more with the communication style of the leaders. All this paved the way to the arrival of the third wave today. The charismatic gifts of the Spirit have now become the power of communication of a charismatic leadership. The competition is an adaptation to the pure logic of the market, governed by the rules of supply and demand, where salvation goods—symbolic products designed for immediate consumption that are aesthetically attractive, and are designed to produce a sense of marvel (somewhere between entertainment and dramaturgy)—circulate freely.

## 2. The Research Questions

Where does the Catholic Charismatic Movement fit into the global Pentecostal landscape? Since 1967, it has moved from the imagination of a pneumatic church to the canonical status of an Ecclesial movement recognized by the Catholic authority. The brief description of the case study gives a chance to make some more general comments on what is happening under the sacred canopy of Global Catholicism where the Spirit blows, and furthermore in relation with the so-called Global Pentecostalism. In other words, the working hypothesis which I intend to discuss includes the following statements:

(a) Catholic Pentecostalism constitutes a variant of a more global phenomenon, which seems to challenge the organizational model of historic Christian churches.

(b) The study of the Italian case is interesting because its story shows the extent to which Pentecostalism questions the Roman form of Catholicism. In some other countries with a Catholic majority—like Brazil, the Philippines and Poland—the development of the phenomenon has not encountered the same difficulties as in Italy. In many cases, the founders and promoters were friars or priests. In Italy, in early seventies, at the beginning of the charismatic movement, there were lay people and some priests united by the desire to experience the Baptism with the Spirit. They met by choice not in the parishes but in private homes, or in houses where some members lived in common. Hence, they received a certain initial distrust on the part of parish priests and bishops of the Dioceses. They saw in this movement an alarming sectarian tendency and worrying approach to Protestantism. The Critical attitude of the ecclesiastical hierarchy intensified in the face of the tension which arose in the late seventies within the charismatic movement. The tension polarized two groups: those who felt they were the prophetic vanguard of the 'third church' (the Church of the Holy Spirit's age) and those who thought that their experience was a sign of renewal of the Catholic Church, not its overcoming (Pace 1983). In Italy, the RnS movement was gradually brought back to the bed of ecclesial orthodoxy, after a long persuasive work carried out by bishops and theologians towards the leaders of the movement itself. The compromise occurred when the discernment of the spirit became the fundamental criterion for distinguishing the charismatic effervescence without limits from an individual and communitarian spiritual experience of the gift of the Holy Spirit, obedient to the indications of the Catholic hierarchy. However, despite this ecclesification/clericalization process, the charismatic tension remains, and the expectation for a pneumatic church constitutes an implicit form of criticism of the Roman form of Catholicism.

## 3. Data and Methods

The data used in this article come from a series of research that the author conducted in various moments of his (not short) scientific activity on different objects. There are four main sources from which the data come:

(a)  A first research carried out in 1977–1980 as part of a project financed by the Italian Ministry of Education and University on ecclesial movements born in the post-Vatican II Council; part of this investigation was dedicated to the reconstruction of the origins of the Catholic charismatic movement (from 1970 to 1980). The results can be found in Pace (1978, 1983, 2006).

(b)　A second research carried out in 2008–2011, funded by the Cassa di Risparmio Foundation, on the diffusion of the neo-Pentecostal churches of Nigeria and Ghana in Italy, conducted on a representative sample of pastors of these churches, with the construction of a map of the their installation, and with two study stays of one month in Ghana and Nigeria conducting interviews and ethnographic observation on the phenomenon; the results can be found in Pace and Butticci (2010), Pace (2013), Butticci (2016).

(c)　A third investigation in 2013–2014, on behalf of the Treccani Institute of Rome, on the presence in the Italian regions of some Catholic ecclesial movements (neocatechumenal, CCR, Communion and Liberation) born in the post-Vatican II period, mapping their diffusion in the territory and recording the main social activities that each of these movements has promoted in the different regional realities; the results can be found in Pace and Contiero (2015).

(d)　A recent survey, conducted in 2019–2020 on behalf of the World Religions and Spirituality Project (WRSP, directed by David Bromley, Virginia Commwealth University), on the origins and spread of the Assemblies of God in Italy.

The third research (2013 and 2014) included a cartographic survey on the territorial diffusion of the CCR Italian affiliated groups that adhere to the ICCRS (International Catholic Charismatic Renewal Service (but now, since 2018, CHARIS, the Catholic Charismatic Renewal International Service)).

The religious geography of Italy has indeed changed. Italian society has historically been influenced by Catholicism in its Roman form. Between the Sixties and the Eighties, Catholicism first experienced an internal differentiation, a singular pluralism (Garelli et al. 2003); all Catholic, but with many different orientations, not only political and social, but also in the theological field and in the way of being a church, sometimes overcoming the historical separation of clergy and laity. Later, with the arrival of immigrants from more than 180 different countries in the world, in a few years (over a decade), Italian society discovered that it was inhabited by Muslims, Sikhs, Hindus, Orthodox and the many enthusiastic faithful affiliated to the new African, Asian and Latin American Pentecostalism. The map concerned the places of worship and aggregation centers of the new religious groups, both those formed within or on the margins of the Catholic Church and in relation to the different confessions and congregations that immigrants brought with them (Pace 2018a).

To map the CCR communities, a first directory was established, consulting the official site of the movement. Where there were no reliable or official sources of the groups themselves, we proceeded with a field research, consulting stakeholders (parish priests and leaders of Catholic associations). In the end, it was possible to have a simple database where we collected basic information (addresses, complete with postal codes; dates of foundation; and the name of the leader or representative). We processed the data relating to the addresses with the TargetMap software program by region and province (Schiavinato 2013).

## 4. CCR in Italy and in the Global Pentecostalism

The Catholic Pentecostals in Italy are an interesting case to study for at least two reasons. First, because the phenomenon was born and developed *near Rome* (in the sense of proximity to the center of world Catholicism); and secondly because it is one of the various movements of spiritual awakening born in the historical climate of the Second Vatican Council (Diotallevi 1999; Melloni 2003; Faggioli 2016; Sprows-Cumming et al. 2019). Catholic Pentecostalism thus expresses a *born-again need* that the Council somehow voiced, a need that went partly unheard or took a completely different direction, i.e., a call for a reform of the Catholic Church.

Renewal in the Spirit (RnS) (this is the official name of the movement today) is a variation on the born-again theme that can bring together people who feel the need for an inner conversion within the Catholic Church. Their need is nourished by an idea of the primitive Christian community, and especially that of Pentecost. Many pro-Vatican-II Catholics were convinced that this would involve renewing the Church without wishing to subvert its constitutive principles. The RnS initially shared the same wish that other movements—some of which, like Chiara Lubich's Focolari, were born well before

the Second Vatican Council (Callebaut 2017)—succeeded in expressing in their various ways. Its stance was in contrast with those fighting for a systemic reform of the Church, like the Neo-Catechumenal movement, or the Cursillos of Christianity, and the many spontaneously established Biblical studies groups (Pace 1983).

Given the centrality of the baptism of the Spirit, it is worth comparing the RnS movement with what happens outside Catholicism in the various forms of world Pentecostalism. For example, the need to be born again that is met by the baptism of the Spirit is intercepted in Italy today (as in other countries around the world, for that matter) not only by the RnS but also by a network of congregations that refer mainly to the Assemblies of God. This historical organization became established in the USA in the 1920s, and soon spread to Italy in 1930. The missionary work of two Italo-Americans (Lombardi and Francescon) met with some success in the south of the country (Stretti 1998; Naso 2013). Since the early 1980s, there has also been a growth in the presence of many diverse Neo-Pentecostal Churches that immigrants have brought with them to Italy (Pace and Butticci 2010).

The Catholic Church only recognized the RnS officially in 1996, twenty years after it was born. It was only after a long journey, in fact, that the Catholic episcopate's initial mistrust of the movement gradually dissolved. Today, the RnS is well established all over the country, with about 250,000 people, organized into a total of 1780 small groups and communities (Figure 1).

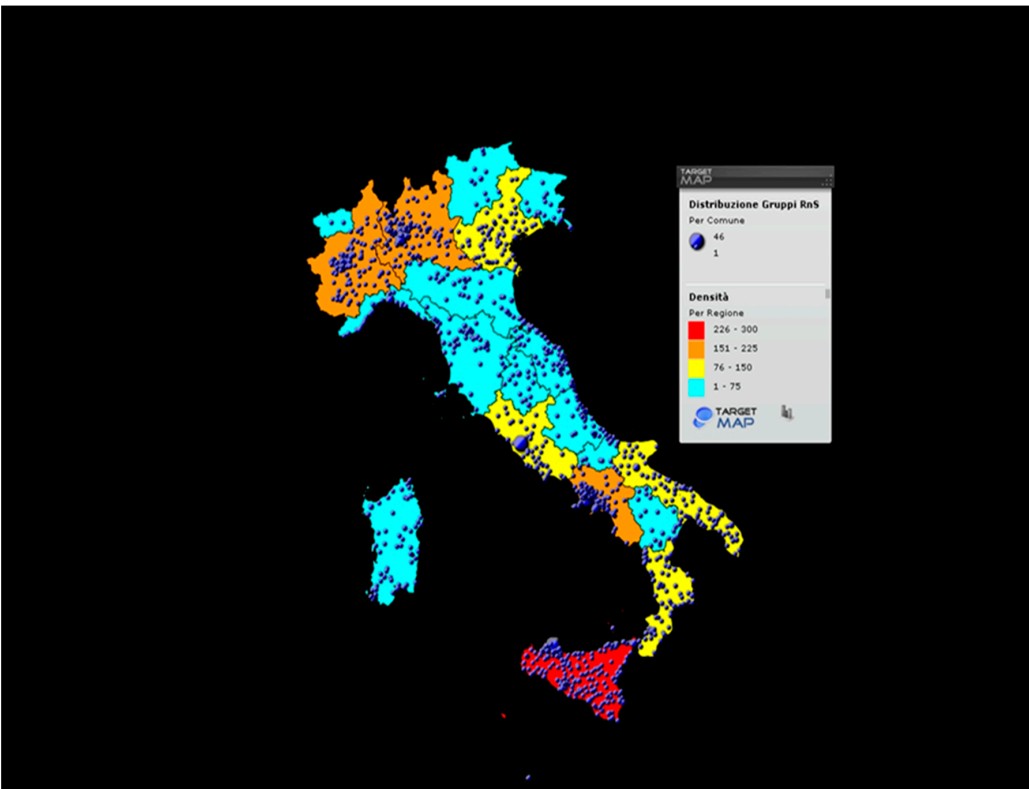

**Figure 1.** The RnS communities in Italy.

Looking at the diffusion of the RnS and of the other two non-Catholic charismatic groups and churches—the 1182 Assemblies of God (Figure 2), and the 650 Nigerian and Ghanaian Neo-Pentecostal Churches (Figure 3)—we can see that they share a demand for spirituality that goes beyond denominational barriers and national borders. It is also worth noting that, through migratory flows, the first vanguards of the new Latin American Pentecostal churches have also arrived in Italy, like the Brazilian Igreja Universal do Reino de Deus (with eleven locations in Italy), and several other similar Churches coming from Central America and China. Moreover, remaining within the Catholic

environment, there is also the Catholic charismatic movement of Mike Velarde's El-Shaddai (which spread with the vast Philippine diaspora also present in Italy).

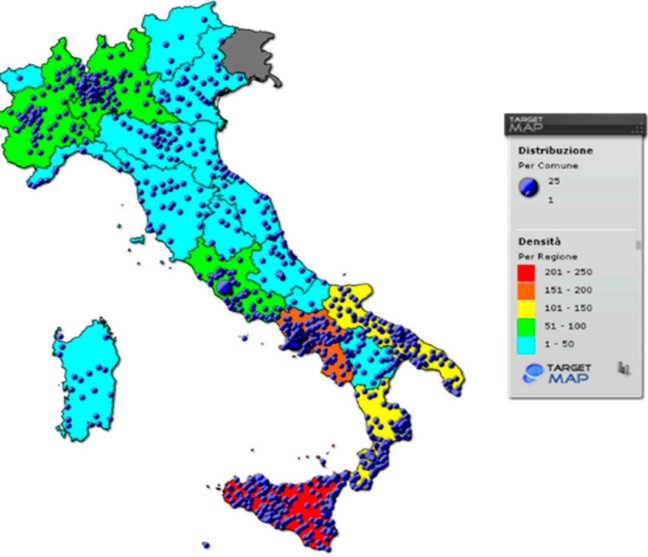

**Figure 2.** Assemblies of God in Italy.

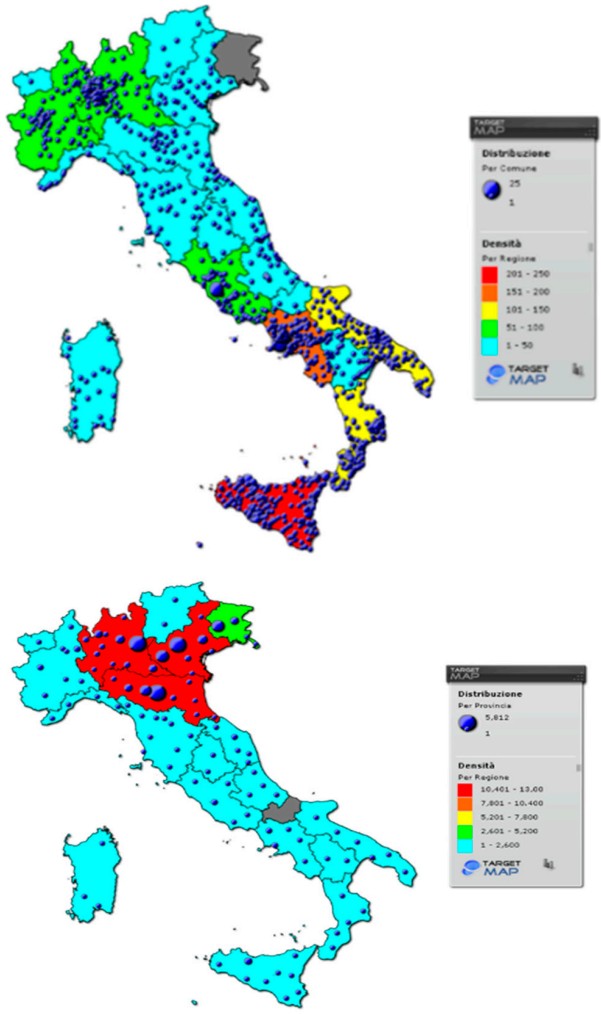

**Figure 3.** Neo-Pentecostal Churches from Nigeria and Ghana in Italy.

Comparing the maps of CCR and the Assemblies of God, what is striking is the success of the Pentecostalist pattern of religiosity, especially in Southern Italy (Campania, Calabria, Puglia and Sicily). In these historically Catholic areas, still steeped in the traditions and devotions typical of popular religion, increasing proportions of those born into Catholicism have been attracted to Pentecostalism, and abandoned Catholicism in favor of the Assemblies of God or the Pentecostal Churches (Berzano and Introvigne 1994; Introvigne and Zoccatelli 2010; Pace and Contiero 2015). In other words, the spread of both Catholic charismatic communities and classical Pentecostalism among Italians, beyond theological and organizational differences, shows that there is a large and articulated Pentecostal religious field, whose borders are not so clear as to allow a mobility of people who, born Catholic, former members of CCR groups, become members of Pentecostal formations that appear to them in continuity with the previous experience but, at the same time, freer, without the control of a clerical hierarchical apparatus anymore. Under a Catholic skin, an exuberant spirituality circulates outside the traditional spaces of the Catholic parishes. Though different, this spirituality appears to be contiguous with the parallel forms of non-Catholic Pentecostalism that are gaining ground, especially in some of the Southern Italian regions, and Sicily in particular.

### 4.1. The CCR in Italy: The Beginning

The CCR is the vessel officially recognized by the Catholic Church into which the flow of Catholic Charismatic Renewal (CCR) has been channeled here in Italy. The CCR and the RnS are therefore not two different things; the latter is a peculiar implementation of the former that has taken root in Italy. Historically, the CCR owes its origin to the Protestant Pentecostal movement. In fact, it was born in America in 1967, just after the end of Vatican II, as a result of an encounter between some Catholic intellectuals from Duquesne University of Pittsburgh, Pennsylvania and some exponents of classical Pentecostalism.

The officially acknowledged story goes that the event marking the birth of the movement took place on 17 February 1967, at the Duquesne University in Pittsburg, when a group of people (about thirty university students and professors) had a direct experience of the presence of the Holy Spirit during a spiritual retreat. When the news of these events began to spread from the university campuses to the parishes, the Jesus Restoration Center (JRC) reached first the United States and Canada, and later Latin America and Europe. It continued to spread further, to other continents, where it gradually took on various forms of organization and canonical status, depending on the peculiarities of each nation. It is now a vast movement disseminated in 204 countries on five continents, with a membership of over a hundred million Catholics.

In Italy, the Duquesne's effect first reached Rome between the end of the 1960s and the early 1970s. Father Valeriano Gaudet, a Canadian missionary priest, accompanied by the daughter of Canada's ambassador to Italy, Jaqueline, and her Italian husband, Alfredo Ancilotti, had a crucial role in promoting the movement on Italian soil. According to a commonly held view, Carlo Maria Martini (the cardinal of Milan) supported the movement. Coming back from the United States, where he had come into direct contact with the movement, he did his best to accredit its position in Italy, starting from within the pontifical Gregorian University in Rome. In fact, it was here that the first charismatic prayer group in Italy, the 'Lumen Christi', was established. This was an international and ecumenical group consisting largely of students, priests and lay people who communicated with each other mainly in English. Over time, other French and Spanish groups emerged, followed by some Italian ones like the 'Maria' group led by Ancilotti and his wife. This group was originally known as *Emanuele*, but its name was changed to *Maria* to emphasize the Catholic inclinations of its adherents. This was the first nucleus—which therefore was called the 'mother group'—from which other groups went to settle in the main Italian cities over the years (Roldan 2009).

A general diffidence on the part of the church hierarchies towards the charismatic phenomenon meant that it struggled a little to take root in Italy. At first, the movement seemed to have settled on positions that were too critical of the church as an institution, which was thought to be in a state of

'crisis' and in need of renewal (Pace 1983). The primacy of the Spirit seemed to put the institutional role of the church in the background, and in brackets, reducing the classic distinctions between clergy and lay people, with the latter usually relegated to minor, passive roles.

In 1976 there were strong internal tensions with the *Emanuele's* group concerning the leadership (Roldan 2009), which led to conflict between the promoters of the movement. On the one hand, there were Ancilotti and his wife, who proposed to transform the figure of the founders into a hierarchical authority so as not to lose the strong lay identity that characterized the movement. On the other, there were those who wanted a more collegial organizational approach, rather than being guided by a single leader. Their belief in the extraordinary powers of the Spirit had led them to think that there was not really a 'founder', as it was only the Spirit that had the power to initiate the charismatic experience. The latter group was also in favor of a greater participation of the clergy, who were also in positions of responsibility within the group. The main argument against the Ancilotti couple's supporters concerned their *unregulated* use of the charisma, as they rejected any spiritual authority or guidance from priests or other religious figures. The more *orthodox* fringe of the movement thus began to implement a gradual process of rapprochement with the Catholic hierarchy. The group's attitude openly changed from being *critical* to accepting a spiritual guides suggested by the Catholic hierarchy, such as the theologian Grasso or Cardinal Suenens.

The outcome of this controversy was a gradual softening of the conflicting positions, leading to the movement gaining more credit with the Vatican authorities. To clearly emphasize the Catholic identity of the movement, the leader decided to call the Italian Catholic Charismatic Renewal movement 'Renewal in the Holy Spirit'. They were inspired by a letter from Saint Paul to Tito (3, 5), in which the apostle wrote, "We are saved through a washing of regeneration and renewal in the Holy Spirit". This decision also eliminated any ambiguity deriving from the use of the adjective 'charismatic', which carried a risk of focusing too much attention on the extraordinary nature and monopoly of the gifts listed by St. Paul in his Letters, and on the greatness of their Donor. Salvatore Martinez, the current national coordinator of the movement, explained that "It was easier to remember that no-one can conveniently be charismatic if not in reference to the Church, because it is charismatic" (Martinez 2015).

This marked a turning point which accelerated the definitive split in the previous movement. A new Catholic charismatic group was born, first on the margins and then outside the official Church, partly swelling the ranks of the Assemblies of God. In the early 1990s (after the split), the General Secretary of the Italian Episcopal Conference (CEI) informed the leader of RnS of the opportunity to initiate the procedure for having the movement recognized according to the Code of Canon Law. The CEI's Permanent Episcopal Council approved the statute between 22–25 January 1996. Subsequently, in 2002, a new statutory text was adopted *ad experimentum* for three years. It was not intended to be in lieu of the previous statute, but only an updated version of the same. The CEI finally approved the new statute in 2007, at the end of the trial period imposed to allow the RnS to consolidate its new course and, at the same time, to verify the real usefulness of the changes it had introduced.

The present statute governs all aspects relating to the movement's nature, aims, activities and relations with bishops. It also establishes the main lines of its spirituality (its practices and theological definition), its training methods and any other practical organizational issues. It thus sanctions the complete transition from a *spontaneous current of grace* to an *ecclesial movement*. The Vademecum of the Movement stresses that "this passage has led to the awareness that a norm not only does not imprison the Spirit, but rather assures each and every one a different and more mature freedom, and the Movement's gradual evolution towards its deepest Catholic and ecclesial identity" (Vademecum Rns 2007–2010, p. 21).

### 4.2. The Story of the Italian CCR in Comparison with Other Case Studies

We cannot generalize from the case of the Catholic charismatic movement in Italy. In other countries, the movement met less resistance from the local Catholic hierarchy and parish priests, or was promptly accepted by them. In some countries they came to appreciate quickly the contribution that

the women and men of the movement could offer to the pastoral activities of a parish—not only for those traditionally entrusted to the laity, but also for some important diaconal functions. Priests caring for large parishes willingly delegate some of these functions, possibly adapting them to practical and urgent needs that arise from time to time.

This is the case in Brazil. Two Jesuits, Haroldo Rahm and Eduardo Dougherty, founded the CCR in Campinas (São Paulo State) in 1969, two years after the 'Duquesne meeting'. The movement expanded throughout Brazil, especially in the first half of the 1990s, competing with the new charismatic mega-churches like the Universal Church of the Kingdom of God (IURD) (Prandi 1998; Corten 1999; Mariz 2004; Correa 2013; De Alencar Freire 2013). The strong expansion of Neo-Pentecostalism—with its consolidation of models of church-enterprise founded and guided by charismatic leaders capable of challenging both the Catholic Church and the historical Churches of the Reformation—has generated some mimetic effects in the Catholic field as well (Oliveira 1978; Oro 1996; Carranza 2000, 2011; Ferreira de Araújo 2019). One of the terrains where competition is most lively is, precisely, music. The CCR distinguished itself in Brazil by taking care of this dimension to make proselytes. One of the best-known priest-singers is Father Marcelo Rossi, but there are others of lesser resonance. What emerges is the fact that some Catholic priests have decided to adopt styles of communication during mass that are not unlike those of their competitors, the charismatic leaders of the new Pentecostal churches.

In France, the Emmanuel Community (the name of the French CCR), founded in 1973 by two ley Catholics, built a global network in ten years, from the local parishes and national dioceses to international organizations like the ICCRS (since 2018, CHARIS). The community tried to be accredited as a group of Catholics actively engaged in the parishes in a deeply secularized France, coping with the initial distrust on the part of the French bishops. John Paul II's papacy marked the leap in quality of Emmanuel. They started to cooperate directly with Rome when the Polish Pope launched the World Youth Days in 1985. In the event supervised by the Pontifical Council for the Laity (PCL), the Emmanuel community animated the Palm Sunday Mass. From this event, the Emmanuel appeared to the PCL as a community devoted to the Pope they could rely on. In 1992, the PCL recognized the Emmanuel Community as an international association of the faithful of the Pontifical Right (Mercier 2019).

In Poland, the Catholic priest Franciszek Blachnicki (1921–1987) founded the Light-Life movement in 1976, two years before the election of Karol Wojtyla and the political and social revolution against the communist regime. The movement combined the traditional Marian spirituality and devotion, rooted in Polish Catholicism, with the charismatic experience. The founder believed that the new theology coming from the Vatican II (in particular, the 'Gaudium et Spes' document) legitimized the charismatic renewal (Siekierski 2012; Hocken 2004). In 1998, John Paul II declared in an official meeting with the national representatives of various CCR that their experience was one of the many fruits of the Vatican II: "like a Pentecost, leading to an extraordinary flourishing in the Church" (Paul 1998, p. 4). It is not difficult to think that John Paul II came to this solemn recognition thanks to the experience of the Light and Life movement which, since its origin, has shown absolute dedication to the Catholic Church.

The American influence in the spring of the CCR both in the UK and Australia was illustrated by Maiden (2019) and Rocha et al. (2020). In particular, the founders of the British and Australian movements, at the beginning of the seventies shared the same ecumenical spirit that favored the first Catholic Charismatic community immediately after the meeting they had with the Pentecostals at Duquesne University. The impact of Pentecostalism—from the USA to the UK and Australia—on the various small groups of Catholic priests or friars (it is the case of Britain, where a Dominican and a Jesuit promoted the first Catholic Charismatic aggregation in 1971–1972) defined from the outset the characteristics of the CCR: to be an authentic outcome of the ecumenical dialogue encouraged by the Vatican II. Therefore, in these countries, the development of the CCR was quiet, without any conflicts with the respective national Catholic hierarchies.

A different story concerns the growth of the El Shaddai movement in the Philippines and its diffusion, via the Philippine diaspora, around the world, to Europe—including Italy, of course—and the Americas, Africa and the rich Gulf countries (Wiegele 2005, 2006). The full name of the movement

is El Shaddai (Almighty God in Hebrew) Prayer Partners Fellowship International. Its undisputed leader is Mike Velarde, who once owned a real-estate agency. He became a Catholic evangelical-style preacher in 1982, when he started a radio program that soon gained a wide audience. The turning point in his career as a charismatic leader came when, once a month, he started calling together his faithful listeners for an outdoor prayer in a space near the radio station. When the crowd became too big for the space available, Velarde began to rent more spacious sites (from large stadiums to congress halls), until he found suitable accommodation in San Dionisio (Parañaque) in 2009. His new headquarters, costing about $21 million, are called the House of Prayer, a mega-church–like structure located in Amyel Business Park. The facility covers 10,000 square meters, with 16,000 seats plus standing room for another 25,000 people. The temple was inaugurated by the former President of the Philippines, Gloria Macapagal Arroyo. Velarde is now over seventy. According to data released by the movement, El Shaddai now has about eight million members, plus two million worldwide. Despite some controversies of a political nature (Velarde was accused of supporting President Fidel Ramos in 1992), or linked to his ownership of a TV broadcasting chain (which prompted a dispute between Velarde and Eddie Villanueva, leader of another charismatic movement of evangelical origin, called Jesus is Lord), Velarde can count on the benevolent approval of the bishop of the diocese of Parañaque.

His competitor, Villanueva, in addition to being the undisputed leader of his own movement, was also a candidate for the presidential elections in 2004 and 2010, who then stood as a senator in the 2013 elections. He is a leading exponent of the Bangon Philipinas ('Wake Up, Philippines') party, founded in 2004 as a secular arm of the religious movement. His popularity increased through a tele-evangelical channel (ZOE Broadcasting Network). Though Eddie Villanueva failed to get elected, two of his sons came to sit in the Senate. Before founding the Jesus is Lord movement in 1978, Villanueva had been a professor of economics and finance, the manager of a commercial company and, under Marcos's dictatorship, a militant in an opposition group near left-wing circles. His movement currently claims to have over five million adherents scattered over eighteen districts in the capital, Manila, and eighty provinces in the Philippines, plus a presence in 60 different countries around the world, through the various diasporas.

These brief accounts of two figures like Mike Velarde and Eddie Villanueva probably suffice to show that there is a competition underway between Catholics and Evangelicals on the charismatic religious stage in the Philippines today. The difference between them concerns how the two leaders move in the political sphere. While Villanueva was directly involved in electoral competitions, Velarde initially tried to exploit his popularity as a religious leader in the political arena, but then chose to comply with the directives of the Filipino episcopate and avoid getting involved in the controversial political events that have characterized the Philippines in the last thirty years. It should be noted that the El Shaddai movement was born in 1984, six years after the one founded by Villaneuva. Since the competition between the two movements started, Velarde has gradually adopted Villanueva's communication styles and content. Both preach the gospel of prosperity. Both recover rituals typical of popular religious traditions, inbetween the magical and the sacred, turning simple objects into talismans.

The studies cited above referring brieflyto some national cases show that the CCR is one and multiple, with different stories according to national contexts, transnational dynamics and close confrontation with different forms of Pentecostalism, from the *classic* one up to the *modernist form* of market-oriented charismatic enterprise. The Italian case, therefore, has its peculiarity, which can be summed up as follows: (a) the religious monopoly regime in which the Catholic Church manages the new charismatic movement; (b) the initial distrust of most of the clergy and bishops towards the conflict within the movement, and consequently the spilt of a schismatic group, and subsequently the visible growth of the Assemblies of God, recruiting many newcomers among Catholics led the Italian bishops and the Pontifical Council for Laity to slow down the official recognition of the CCR.

## 5. CCR in the Pentecostal Religious Field

What I have said so far strengthens the conviction that what we conventionally call Pentecostalism is the sign of a process of change that has been accelerating around the world over the past thirty years (Martin 1993, 2001; Cox 1995). Some may have vaguely sensed what was coming years ago, but even those who did certainly could not have imagined the complexity and magnitude of the phenomenon. This change concerns contemporary Christianity, including Catholicism, which saw the Charismatic movement flourish inside. Moreover, the new forms of Pentecostalism increasingly challenge Catholicism, especially in the Southern world.

The magnitude of this phenomenon can evoke (Revees 1980; Althouse and Waddel 2010; Oliverio 2012) what Joachim of Fiore wrote of in his eschatological interpretation of the history of Christianity being punctuated by the passage from the age of the Father to that of the Son, and then to that of the Holy Spirit: the Third Church. Taubes (1994)—the rabbi, philosopher and sociologist of religion who left us such a fundamental work as *Western Eschatology*, his PhD Dissertation defended in Zürich in 1947—reflected at length on Joachim's thought. Taubes revealed its historical and social potential, and had already noticed that the eruption of a theology of the Spirit is a potentially revolutionary event in the history of Christianity. It even goes beyond Christianity, if we consider Ernst Bloch's theory (Bloch 1995) on the hope principle, or on the liberating force of the theology of hope. Indeed, Taubes noted that charismatic movements question themselves, sometimes furiously shouting their doubts: if we firmly believe in the saving power of the Spirit, what is a church doing about it? The power of the Spirit generates human empowerment. Here lies the problem posed by modern global Pentecostalism, not only in the sphere of theology, but also and especially for the sociology of the spirituality (Giordan and Pace 2012). Many Pentecostals think that an organization of salvation like the church is past its time, overtaken by that of the Spirit. Let me quote a short passage from Taubes (1994, p. 117):

> The prophecy of the Kingdom, like *ecclesia spiritualis*, undermines a closed (immobile) system of religious belief that has placed itself at the center, repressing any dissenting or subversive force. The inner light of the *ecclesia spiritualis* reduces the outer walls of the institutions of salvation to ashes. The announcement of the Kingdom incites its realization. The alternating rhythm between spiritual ecclesia and realization on earth characterizes the eschatological movements ... Each new eschatology puts in crisis the dominant sense horizon. Emerges as a new voice ... A new syntax is created and the interruption of meaning that is produced within the language (of a dominant religion) makes the ancient man appear crazy to the new one and vice versa ... The eschatological thought of Joachim of Fiore creates a rift within the system of the city of medieval God. The slogan of the *ecclesia spiritualis* destroys the identification of the church with the Kingdom of God ... The theology of history played out is brought to completion by the theology of revolution by Thomas Münzer, who wants to realize the Kingdom on Earth.

I have chosen to recall the words of Taubes because it seems to me that what we see today in the religious field under the sacred canopy of Pentecostalism, including the *moderate form* within Catholicism, is precisely a set of movements, born outside or inside the historical churches, that implicitly or explicitly negotiate any form of salvific mediation (more or less organized and institutionalized). The church-model appears to have become obsolete because the Spirit reveals itself with its power here and now, which saves and guarantees success in life. All this can be experienced by those who believe in it, who unite together with others in prayer and rely on the oracular word of the charismatic leader.

Therefore, Catholic Charismatic movements belong to a unique (despite internal differentiation) religious field: Global Pentecostalism (Freston 2002; Corten et al. 2005; Meyer 2009; Garcia-Ruiz and Michel 2011; Da Silva Moreira and Mariano 2012; Lucà Trombetta 2013; Da Silva Moreira and Trombetta 2015; Fer and Malogne-Fer 2016). The Bourdieu (1971) concept of the religious field, applied to contemporary Global Pentecostalism, works as a large religious umbrella where the CCR can be located despite its peculiarities as a movement born in a Catholic environment (Vian 2016; Ciciliot 2019).

In the CCR, the struggle to control the means of salvation, and the symbolic capital that this control has accumulated over time, is implicit, because the eschatological tension to *ecclesia spiritualis* does not call into question the pastoral power of ecclesiastical authorities (bishops and the Pope). The more it takes place in the context of a monopoly, the more the control is institutionally placed in the hands of a class of professionals of the sacred. Consequently, there is here a measurable difference between the Catholic Pentecostal movement and the Pentecostals born in the Reformation's matrix, and especially in the third wave of post-Protestant Pentecostalism, which I call, to mark the difference with classical Pentecostalism, the market-oriented charismatic enterprise. While the latter refused any idea of an institutional mediation, the former agreed to be scrutinized by theological *customs officers*. Catholic bishops worried about a forward flight of such movements, which some theologians sometimes accused of becoming like the Protestants. Catholic charismatics and classic Pentecostals, indeed, share an individualized need for *born-again spirituality* (conversion or metanoia) and the conviction that belonging to a Church—by birth and tradition, territorial proximity, cultural and historical roots—is superseded by the strength of the Spirit, which blows where it will. The Pentecostals compared to charismatic Catholics explicitly not only do not recognize themselves in the salvific mediating function of a church-type institution, in line with the ideas originating from the Reformation, but they try to create forms of assembly or congregational-type organizations that exalt the equality of all believers before the power of the Spirit. The third wave of Pentecostalism (the 'third church') poses a challenge both for the historical Pentecostal congregations born in the Protestant environment and for the Pentecostal movements within the Catholic Church. The new churches (established since 1980) carry the dual contingency characterizing Pentecostalism to its extreme consequences. The charisma becomes a transnational enterprise (as in the new churches in Latin America, Africa and Asia) (Pace 2017). The leaders of these churches adopt market-oriented messages and forms of organization. They act as advertisers of salvation commodities in the salvation goods market. The ritual becomes a performance, and the believer a loyal customer for miracles, as well as Christian rock, beautiful songs and mass catharsis. Please see Figure 4.

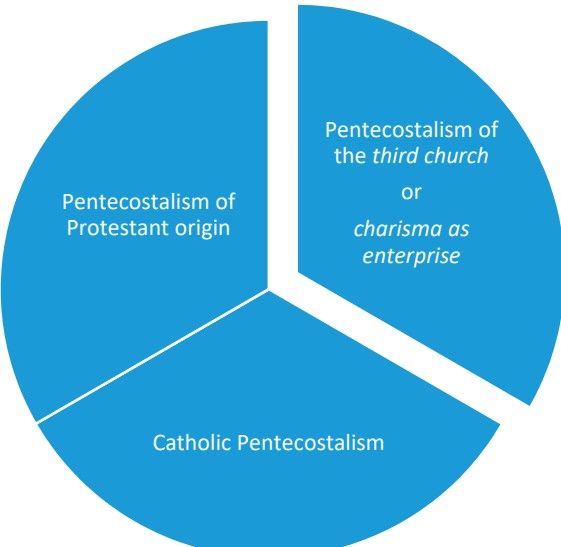

**Figure 4.** The differences within the Pentecostal religious camp.

## 6. Global Pentecostal Catholicism

Going beyond their differences, it is worth stressing the new configuration of the Christian religious field worldwide produced by Global Pentecostalism (Freston 2002; Corten et al. 2005; Meyer 2009; Garcia-Ruiz and Michel 2011; Da Silva Moreira and Mariano 2012; Lucà Trombetta 2013; Da Silva Moreira and Trombetta 2015; Fer and Malogne-Fer 2016), in which I deliberately include the network of Catholic charismatic movements. Global Catholicism is a part of Global Christianity (Vian

2016), meaning that the vital presence of such movements signals a change in the way of believing and belonging, even in and to a *church of birth* (Pace 2018b).

The story of the Pentecostal movement in Italy seems like a sort of laboratory test demonstrating my hypothesis. The long negotiations that finally led the movement to be fully recognised by the ecclesiastical authorities show to what extent the third type of socio-religious organization studied by Troeltsch (1947) objectively represents an intolerable challenge for any institution of salvation claiming to exercise its function of sacred mediation as a monopoly. In organizational terms, the *spiritualismus* (Troeltsch's third ideal type) in Catholicism has been a way of practicing religious faith outside clerical control, closing the gap between clergy and laity. It was potentially in antithesis to the organizing principle established by the Council of Trent: to bind believing and belonging closely together through the virtue of obedience to the authority of the hierarchy. If we read the story of Catholic Pentecostalism, taking this long-term perspective, we can see why and how it is a way of believing potentially at odds with the standardized model controlled by a religious knowledge and power that are monopolized by the clergy.

I would like to provide another example as an extreme test of this assumption by referring to the case of Milingo (ter Haar and Ellis 1988; ter Haar 1992; Csordas 2007). The conflict within the Roman Catholic Church is evident from the story of the Zambian Archbishop Emmanuel Milingo in the years from 2001 to 2009. Milingo credits himself with being a charismatic leader capable of interceding with the Holy Spirit, liberating with exorcism, and healing psychic and physical ills. He was a sort of freelance Pentecostal who set up on his own and an official representative of the Roman Church at one and the same time. The following that Milingo gained within a few years alarmed the Vatican authorities, who invited him to stop organizing public gatherings. Instead, Milingo emphasized his profile as a charismatic entrepreneur who does not feel compelled to obey orders from Rome. His resounding involvement with the Church of Reverend Moon, and the wedding celebrated according to the rules of this South Korean church with a young Moonist adept, resulted in him being reduced to the lay state in 2009 for repeatedly challenging the rules of canon law (ordaining priests and bishops).

Another significant example worth mentioning concerns the growth of the El Shaddai movement in the Philippines and its diffusion, via the Philippine diaspora, around the world, to Europe—including Italy, of course—and the Americas, Africa and the rich Gulf countries employing laborers from the Philippines (Wiegele 2005).

The full name of the movement is El Shaddai (Almighty God in Hebrew) Prayer Partners Fellowship International. Its undisputed leader is Mike Velarde, who once owned a real-estate agency. He became a Catholic-evangelical-style preacher in 1982, when he started a radio program that soon gained a wide audience. The turning point in his career as a charismatic leader came when, once a month, he started calling together his faithful listeners for an outdoor prayer in a space near the radio station. When the crowd became too big for the space available, Velarde began to rent more spacious sites (from large stadiums to congress halls), until he found suitable accommodation in San Dionisio (Parañaque) in 2009. His new headquarters, costing about $21 million, are called the House of Prayer, a mega-church-like structure located in Amyel Business Park. The facility covers 10,000 square meters, with 16,000 seats plus standing room for another 25,000 people. The temple was inaugurated by the former President of the Philippines, Gloria Macapagal Arroyo. Velarde is now over seventy. According to data released by the movement, El Shaddai now has about eight million members worldwide. Despite some controversies of a political nature (Velarde was accused of supporting President Fidel Ramos in 1992), or linked to his ownership of a TV broadcasting chain (which prompted a dispute between Velarde and Eddie Villanueva, leader of another charismatic movement of evangelical origin called Jesus is Lord), Velarde can count on the benevolent approval of the bishop of the diocese of Parañaque.

His competitor, Villanueva, in addition to being undisputed leader of his own movement, was also a candidate for the presidential elections in 2004 and 2010, then stood as a senator in the 2013

elections. He is a leading exponent of the Bangon Philipinas ('Wake Up, Philippines') party founded in 2004 as a secular arm of the religious movement. His popularity increased through a tele-evangelical channel (ZOE Broadcasting Network). Though Eddie Villanueva failed to get elected, two of his sons came to sit in the Senate. Before founding the Jesus is Lord movement in 1978, Villanueva had been a professor of economics and finance, manager of a commercial company and, under Marcos's dictatorship, a militant in an opposition group near left-wing circles. His movement currently claims to have over five million adherents scattered over eighteen districts in the capital, Manila, and eighty provinces in the Philippines, plus a presence in 60 different countries around the world, through the various diasporas.

These brief accounts of two figures like Mike Velarde and Eddie Villanueva probably suffice to show that there is a competition underway between Catholics and Evangelicals on the charismatic religious stage in the Philippines today. The difference between them concerns the way in which the two leaders move in the political sphere. While Villanueva was directly involved in electoral competitions, Velarde initially tried to exploit his popularity as a religious leader in the political arena, but then chose to comply with the directives of the Filipino episcopate and avoid getting involved in the controversial political events that have characterized the Philippines in the last thirty years. It should be noted that the El Shaddai movement was born in 1984, six years after the one founded by Villaneuva. Since the competition between the two movements started, Velarde has gradually adopted Villanueva's communication styles and content. Both preach the gospel of prosperity. Both recover rituals typical of popular religious tradition, in-between the magical and the sacred, turning simple objects into talismans.

## 7. Conclusions

The case of the Philippines provides useful pointers to help us understand how a Catholic Charismatic movement ends up resembling a movement of Evangelical inspiration. In doing so, it proves capable of competing effectively with the latter. In the ensuing competition, the Catholic leader increasingly adopts the persona of the transnational entrepreneur of the charisma that his evangelical counterpart has already exploited successfully. The difference between them then lies entirely in the bonds of belonging: Velarde is keen to be recognized as a leader of a movement that remains within the Catholic Church; Villanueva, having no historical religious institution behind him but only the galaxy of new churches of the Spirit, can move freely, enhancing his personal gifts. Confirmation of this situation comes from the two leaders' claims regarding the numbers of their followers. Their figures are impossible to verify. We have no reliable means to ascertain how many people are involved in either movement. Velarde and Villanueva can boast as much as they like, and if one has claimed to have eight million members, it is because he is trying to convince the public that he is doing far better than his competitor, who spoke of having seven million.

In a regime of religious monopoly (as in the Philippines or Italy), Catholic Charismatic movements could emerge and then evolve towards an organizational model that I call 'charisma as an enterprise', providing the ecclesiastical authorities do not intervene and call the movement to order (Butticci 2016; Diotallevi 2018). As in the Italian case, potential competition with other non-Catholic movements can, within limits, be managed by the Catholic hierarchy, and made compatible with the functioning principles of the Catholic institution. But where the Church's monopoly has already been partly eroded by the eruption of a plurality of (first-generation) Pentecostals and Charismatic movements (of the second Pentecostal generation) on the religious stage, a Charismatic Catholic movement finds itself competing with the others to produce the same salvation goods (Pace 2019). Pentecostal movements challenge the exclusivity and universality of the Catholic Church, going against the famous saying wrongly attributed to Cyprian (Bishop of Carthage, in 249–250 AD), *extra ecclesiam nulla salus*, that was firmly emphasized by Pope Benedict XVI in the document 'Dominus Jesus' (September 2000). The Pentecostals would reword it as: 'The Holy Spirit saves . . . *extra ecclesiam*'.

The Catholic Church continues to claim its exclusive mediatory role in the economy of salvation, but in the social and religious worlds of Brazil and the Philippines (and also, in proper proportion, in those of Nigeria, Ghana, the Ivory Coast, Congo, and elsewhere) (Corten 2002; Echtler and Ukah 2015) the Spirit seems to blow outside the boundaries set by the Catholic Magisterium. What does it mean when the leader of a Catholic Pentecostal movement starts imitating the winning methods of a leader of a Neo-Pentecostal church? Or when a Catholic priest or layperson leading a Catholic movement becomes as successful as a preacher as many Evangelical or Neo-Pentecostal leaders?

It could be that the religious message can adopt the logic of modern-style communications in a crowded communicative space. This tends to turn the religious worldview into a religious *promotional* message, conceived to focus increasingly on a market of salvation goods. The leader's risk of failure is higher, and Catholic leaders are aware that their own failures could have a bearing on the institution to which they belong. What essentially distinguishes situations like those seen in Italy or France from the Philippines and Brazil is the bond between collective religious imagination and the personalization of leadership in Pentecostal movements. The *mundus imaginalis* that the theology of the Holy Spirit creates 'lives' in the physical person of the charismatic leader. The extraordinary personal gifts they are believed to possess mediate the power of the Spirit, and this power serves to legitimize their authority as leaders, founded on their oracular and divinatory power.

Pentecostalism is a religious field where different forms of struggle can take place for the control of the means of salvation: from the self-consumption of the goods of salvation individualized and disciplined by the figure of the big man (the charismatic leader) (Kalu 2008, p. 103) up to religious movements animated by an eschatological tension towards the spiritual church.

**Funding:** This research received no external funding.

**Conflicts of Interest:** The author declares no conflict of interest.

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
