# Peer review of "The Catholic Charismatic Movement in Global Pentecostalism"

_religions, doi:10.3390/rel11070351_

Round 1

Reviewer 1 Report

This article has interesting material about the distinctive unfolding of the CCR (Catholic Charismatic Renewal) in Italy, and the abstract suggests that locating Italy’s distinctiveness vis-à-vis the CCR elsewhere in the world will be the focus. The title, however, suggests that the CCR in relationship to global Pentecostalism will be the focus, and the last few pages that discuss Brazil and the Philippines seem to argue for the latter.

            I find the article thus unclear about what its main argument is. Italy’s CCR? Or the CCR in general? The author tries to have it both ways with the two part thesis statement (a and b), but I don’t think it works. No matter which direction the article takes, more work needs to be done and once the decision is made about what argument to make, then the author should remove extraneous material.

            I’d recommend focusing on the Italian CCR history as distinctive in relationship to CCR history and circumstances elsewhere. I think that is interesting and important. The rest of the material is less original and thus less interesting—that is, as a generalization about the breakdown of ecclesial forms globally, with the CCR evidence from within Catholicism of a larger pattern. Of course, if that’s what the author wants to focus on, then he or she could.

            If the Italian case, however, will be the focus, then I think the author should 1) make clear how ICCRS fits into this. Am I right that the former ICCRS has now been replaced by by CHARIS as the global CCR body that the Vatican has organized since Dec. 2018? (I honestly don’t know—just did a quick internet search to find that out). 2) how does this Italian “Renewal in the Spirit” fit into ICCRS/CHARIS if at all? And the key issue: 3) how does Italy’s CCR unfolding compare to the CCR elsewhere?

            I think the limited comparison with Velarde and his El Shaddai and the mention of Milingo, and the references to Brazil’s Pentecostal churches confuse matters, unless the nature of the comparison is expressed more clearly. El Shaddai has had tensions with the Filipino bishops I believe, but is there comparable tension with Italian bishops and the CCR? And is El Shaddai connected formally with the CCR in the Philippines? How do other countries compare to Italy’s experience.

            My recommendation would be to do more with the demographic and cartographic materials presented, explain their relevance better. Why is it important that there are more CCR and Pent’l folks in S. Italy? How does that relate to variations in CCR participation in other countries?

Author Response

Thanks for your comments and advice.
Following them I rewrote the introduction to clarify the perspective I took in the article: moving in concentric circles from the particular (the case of the CCR in Italy) to the general (other examples of the CCR and the relationship with the Global Pentecostalism). This perspective summarizes years of research, which I began in 1977-78 in Italy on the first Pentecostal Catholic communities, the Maria and Alleluja communities, continued later on the neo-Pentecostal Churches in Nigeria and Ghana and on their presence in Italy and ended with a investigation into the origins and roots of the Assemblies of God in Italy. In the middle there have been various study stays in Brazil, Chile, Argentina, USA and South Korea to understand both Global Pentecostalism and the relationship between this phenomenon and the CCR which has now become a global network (within the Catholic Church, but with a degree of relative autonomy).
Therefore, after writing about the CCR in Italy in the past, I consider more interesting to study it as an articulation of a religious field (in the Bourdieu sense) that we conventionally call Global Pentecostalism. If, on the one hand, therefore, I do not feel like following your advice to focus the article only on the Italian case, I have accepted some of your comments. In particular, in addition to rewriting the introduction, where I clarify what I have said, I have also rewritten the paragraph "Data and Methods" (see in red), where I briefly reported the research itinerary completed by providing the sources to which I draw the data summarized in the article; I compacted the other case studies relating to the JRC in a single paragraph, clarifying why I focus with particular attention both on Brazil (due to the mimetic effect of Global Pentecostalism) and on the Philippines (case of tendency to Pentecostalize Catholicism); I clarified the notion of religious field (see in red) and delete the figure 4. 

Reviewer 2 Report

I think it is a very valuable article and I suggest it is published in its present form.

Author Response

Thank you

Reviewer 3 Report

In my opinion, this contribution is fascinating, even with a sociological approach.
The presentation of Global Pentecostalism is sufficiently summarized to introduce Italian implementation.
I would suggest that the authors improve the images. In figure 1, I would remove the black background (or crop the image to reduce it). In figures 1-3, I would eliminate the interpretive panel (practically illegible) and replace it with a text note that associates the colors with the numerical ranges.

On p. 2, lines 61-78 we found the two assumptions of the contribution (a and b).
Hypothesis b is complicated and unclear. "The study of the Italian case is interesting because its story shows the extent to which Pentecostalism questions the Roman form of Catholicism." It seems that the hypothesis coincides with the study.
The hypothesis is reported differently on p. 8, lines 328 and following: "The long negotiations that finally led the movement to by fully recognition by the ecclesiastical authorities show to what extent the third type of socio-religious organization studied by Troeltsch (1947) objectively represents an intolerable challenge for any institution of salvation claiming to exercise its function of sacred mediation as a monopoly". However, in the conclusions, the hypothesis still seems different: "help us understand how a Catholic Charismatic movement ends up resembling a movement of Evangelical inspiration." It would be appropriate to summarize and clearly explain hypothesis b) and then put in a subsequent paragraph how it would be verified or the aspects that make it up.

p. 4: Although I agree with the claim that Pentecostal groups "share a demand for spirituality that goes beyond denominational barriers and national borders", I cannot find support for this hypothesis in the data reported. Apparently, this concept is implicit in belonging to a Pentecostal group. However, the idea of "spiritual" could be very different between groups of American origin (spirituality as a superset of religiosity, and outside institutions) and European ones (spirituality as a subset of religiosity). And this could help explain the Italian transformations to return to the Official Church of Rome (or to adapt to Rome). As lines 174-187 and 204-208 of the contribution would seem to demonstrate.

p. 6: The bibliographic reference to "Vademecum 2007-2010, p.21", contains a phrase which, if retranslated into Italian, is traceable in many web sites. Since it's mentioned a particular page of a document, perhaps it is appropriate to add a bibliographic entry.

Author Response

Thanks for your comments and advice.
Following them I rewrote the introduction to clarify the perspective I took in the article: moving in concentric circles from the particular (the case of the CCR in Italy) to the general (other examples of the CCR and the relationship with the Global Pentecostalism). This perspective summarizes years of research, which began in 1977-78 in Italy on the first Pentecostal Catholic communities, the Maria and Alleluja communities, continued later on the neo-Pentecostal Churches in Nigeria and Ghana and on their presence in Italy and ended with a investigation into the origins and roots of the Assemblies of God in Italy. In the middle there have been various study stays in Brazil, Chile, Argentina and South Korea to understand both Global Pentecostalism and the relationship between this phenomenon and the CCR which has now become a global network (within the Church, but with a degree of relative autonomy).
The main argument: I consider the CCR as one of various socio-religious movements that act in a religious field (in the Bourdieu sense) that we conventionally call Global Pentecostalism. According to your comments, in addition to rewriting the introduction, I have also rewritten the paragraph "Data and Methods", where I briefly reported the research itinerary completed by providing the sources to which I draw the data summarized in the article; I compacted the other case studies relating to the JRC in a single paragraph, clarifying why I focus with particular attention both on Brazil (due to the mimetic effect of Global Pentecostalism) and on the Philippines (case of tendency to Pentecostalize Catholicism); I tried to apply the Taubes's approach (the idea of the "spiritual church") to the various Charismatic movement within contemporary Christianity, showing the different degree of conflict (explicit-implicit) between these movements and the church-type organization. As regard the Vademecum, I added more references.

Reviewer 4 Report

This article gives a very interesting account on the dynamic of global Charismatic movement within Catholic Church and in relation to Global Pentecostalism. It dwells on the interrelation between the Protestant and Catholic framings of Pentecostalism and especially on the external and internal tensions that arise in Catholic Charismatic movement, when it is confronted with different Protestant models and Catholic orthodoxy.

The first hypothesis is that Catholic Pentecostalism is challenging organization model of established Christian churches. Here two most important layers are distinguished. One considers the history of Catholic Charismatic movement in Italy and its formation under the Catholic Church orthodoxy until its recognition within the Catholic Church structures. While this account gives a very interesting insight into the tensions, dialogs and mutual shaping between Catholic Church orthodoxy and charismatic Christian model, some facts and analysis are presented to the reader from the position of all knowing narrator. Therefore the reader is put in front of a readymade narrative without getting access to possible discussions and counter debates that might be at stake of presented processes.

Second layer dwells on the influence of Pentecostal churches that are labeled as “charisma enterprises”, on different other Pentecostal churches, both catholic and protestant. This argument is based in several examples coming from Philippines or Zambia. One grasps easily the idea behind the examples on mimetic processes and capitalization and control of salvation and charisma. However the reader might feel that the argument could be more elaborated. For example Figure 4. Representing differences within the Pentecostal religious camp, that was intended to serve as explanatory graphic on mutual influences of different Pentecostal churches seems not to full fill these expectations. As author writes: “The purpose of the diagram below (Figure 4) is to show how the third wave of Pentecostalism (the “third church”) poses a challenge both for the historical Pentecostal congregations born in the  Protestant environment and for the Pentecostal movements within the Catholic Church” (Line 304-306).

Therefore the theoretical part that stands behind the explanatory model for this layer could be pushed forward by developing the announced application of Bourdieu’s religious field on Pentecostalism, by going beyond the control of salvation and symbolic capitalization of charisma. It might be fruitful to open Bourdieu’s writing beyond the study he made in relation to France. An interesting account on such work might be the application of Bourdieu’s ideas to reflect on religious fields in Africa made by Magnus Echtler and Asonzeh Ukah (Bourdieu in Africa. Exploring the Dynamics of Religious Fields). 

Second hypothesis is connected with Italy, and the author suggests that Italian case illustrates well the situation where the Pentecostal movement is challenged by dominant religious institution. The author suggests that Italian case is unique in relation to other nation states framings. And for comparison the author takes such examples as Brazil and Philippines. As interesting and reflection opening it might be, I would also ask about more comparable cases, especially when looking at the context. Maybe taking in consideration an European and not Non-European cases would be also justified? As the principal key for author’s choices, as one might deduce, was the dominant position of the Catholic Church in particular nation states, there are several cases in Europe that could fit in this category. The choices made by the author are for sure showing the divergencies and building the contrasting narrative, that leads later to relatively slight assumption about the Italian case. From my angle I would look closer on the Polish contemporary dynamic of Catholic Charismatic movement that shows great parallels with authors claims about Italy.  

What is missing are often the sources. At the beginning of the article we get to know that part of the data was collected by mapping religion in Italy project. However the geolocation data and collected metrics do not cover detailed information provided in the article. In many places in the text the sources are not indicated and not explained. The reader does not know if the author was conducting own research on particular institutions (what type of research) or if (s)he worked with particular case studies during own research. Also we do not know any methods and methodologies except geolocation and flat data collecting - metrics.

There are also some details that should be taken in consideration:

  • The references: page 1, line 35 , the surname of one of the co-authors is not Llera but Blanes
  • Birgit Meyer instead of Birgit Mayer, line 323 and 489
  • Line 7 and 8 are not clear: In other words, my working hypothesis which I intend to discuss the following questions: a)…..

What I feel is missing in this article is a solid explanation about the steps of the analytical process and clear track for argumentation. Although the article offers an interesting account on Charismatic Catholic movement, providing many data and comparison with different cases, what could be improved is giving more clarity in developing the main argument, answering the questions and verifying the hypotheses.

Author Response

Thanks for your comments and advice.
Following them I rewrote the introduction to clarify the perspective I took in the article: moving in concentric circles from the particular (the case of the CCR in Italy) to the general (other examples of the CCR and the relationship with the Global Pentecostalism). This perspective summarizes years of research, which began in 1977-78 in Italy on the first Pentecostal Catholic communities, the Maria and Alleluja communities, continued later on the neo-Pentecostal Churches in Nigeria and Ghana and on their presence in Italy and ended with a investigation into the origins and roots of the Assemblies in Italy. In the middle there have been various study stays in Brazil, Chile, Argentina and South Korea to understand both Global Pentecostalism and the relationship between this phenomenon and the CCR which has now become a global network (within the Church, but with a degree of relative autonomy). I consider it more interesting to study it as an articulation of a religious field (in the Bourdieu sense) that we conventionally call Global Pentecostalism. I have accepted some of his comments. In particular, in addition to rewriting the introduction, where I clarify what I have said, explaing the main focus and the main arguments step by step I develop in the text. Morover I have also rewritten the paragraph "Data and Methods", where I briefly reported the research itinerary completed by providing the sources to which I draw the data summarized in the article (All information relating to the narration of the Italian charismatic movement at its origins derives from testimonies collected by the author in 1977-78, confirmed by other researchers who investigated the movement in 2007 and 2010). I compacted the other case studies relating to the JRC in a single paragraph, clarifying why I focus with particular attention both on Brazil (due to the mimetic effect of Global Pentecostalism) and on the Philippines (case of tendency to Pentecostalize Catholicism). I corrected I corrected Blanes and Meyer, inserted Echtler and Ukah's book, and clarified  sentence ex 7-8.

Round 2

Reviewer 1 Report

I think the author has either responded thoughtfully to my comments, or adjusted the paper in ways I recommended. 

Thanks for the work.